# Caffeic Acid Modulates Processes Associated with Intestinal Inflammation

**DOI:** 10.3390/nu13020554

**Published:** 2021-02-08

**Authors:** Danuta Zielińska, Henryk Zieliński, José Moisés Laparra-Llopis, Dorota Szawara-Nowak, Joanna Honke, Juan Antonio Giménez-Bastida

**Affiliations:** 1Department of Chemistry, University of Warmia and Mazury, 10-727 Olsztyn, Poland; dziel@uwm.edu.pl; 2Department of Chemistry and Biodynamics of Food, Institute of Animal Reproduction and Food Research, Polish Academy of Science, 10-748 Olsztyn, Poland; h.zielinski@pan.olsztyn.pl (H.Z.); d.szawara-nowak@pan.olsztyn.pl (D.S.-N.); j.honke@pan.olsztyn.pl (J.H.); 3Group of Molecular Immunonutrition in Cancer, Madrid Institute for Advanced Studies in Food (IMDEA-Food), 28049 Madrid, Spain; moises.laparra@imdea.org; 4Group on Quality, Safety and Bioactivity of Plant Foods, Centro de Edafología y Biología Aplicada del Segura (CSIC), 30100 Murcia, Spain

**Keywords:** inflammation, caffeic acid, antiglycative, antioxidant, myofibroblasts, ACE inhibitory activity, colon, COX-2, PGE_2_

## Abstract

Caffeic acid is one of the most abundant hydroxycinnamic acids in fruits, vegetables, and beverages. This phenolic compound reaches relevant concentrations in the colon (up to 126 µM) where it could come into contact with the intestinal cells and exert its anti-inflammatory effects. The aim of this investigation was to study the capacity of caffeic acid, at plausible concentrations from an in vivo point of view, to modulate mechanisms related to intestinal inflammation. Consequently, we tested the effects of caffeic acid (50–10 µM) on cyclooxygenase (COX)-2 expression and prostaglandin (PG)E_2_, cytokines, and chemokines (IL-8, monocyte chemoattractant protein-1 -MCP-1-, and IL-6) biosynthesis in IL-1β-treated human myofibroblasts of the colon, CCD-18Co. Furthermore, the capacity of caffeic acid to inhibit the angiotensin-converting enzyme (ACE) activity, to hinder advanced glycation end product (AGE) formation, as well as its antioxidant, reducing, and chelating activity were also investigated. Our results showed that (i) caffeic acid targets COX-2 and its product PGE_2_ as well as the biosynthesis of IL-8 in the IL-1β-treated cells and (ii) inhibits AGE formation, which could be related to (iii) the high chelating activity exerted. Low anti-ACE, antioxidant, and reducing capacity of caffeic acid was also observed. These effects of caffeic acid expands our knowledge on anti-inflammatory mechanisms against intestinal inflammation.

## 1. Introduction

Inflammatory bowel disease (IBD) has become an important global health problem, accounting for 6.8 million cases worldwide in 2017 [1]. This elevated number of cases is related to lifestyle choices (i.e., Western diets) characterized by excessive consumption of advanced glycation end products (AGEs), which bind to their intestinal receptors (RAGE) promoting intestinal inflammation [2]. Chronic inflammation of the gastrointestinal tract results in the destruction of the intestinal barrier (i.e., mucosa damage, epithelial erosion, and submucosal edema), thus altering the immune cells–microbiota interaction [3]. This alteration triggers a permanent activation of the intestinal cells characterized by increased receptor expression (such as RAGE and toll-like receptors), increase of oxidative stress, as well as the uncontrolled biosynthesis of molecules involved in the modulation of the inflammatory response, including prostaglandins (i.e., PGE_2_), cytokines (IL-6), chemokines (monocyte chemoattractant protein-1 -MCP-1- and IL-8), components of the renin–angiotensin system (ACE), and adhesion proteins [4]. This chaotic intestinal environment generates a long-lasting inflammatory response favoring the development of ulcerative colitis and Crohn’s disease [3].

The prevention/treatment of IBD encompasses an important number of strategies, including anti-AGE/RAGE treatments [2], modulation of the renin–angiotensin system (i.e., ACE inhibition) [4], attenuation of the inflammatory response targeting intestinal and immune cells, and amelioration of the excessive oxidative stress [2,4]. Conventional anti-inflammatory therapies may result in potential harmful side effects, highlighting the need of new insight into alternative approaches [5]. Dietary phenolics have proven to be effective against a number of chronic diseases, including intestinal inflammation [6]. Hydroxycinnamic acids (Figure 1) constitute a major class of polyphenols widely distributed in fruits, vegetables, and some beverages. Caffeic acid is one of the most abundant hydroxycinnamic acids in the human diet (Figure 1). Potatoes (40 mg caffeic acid/100 g), artichokes (21 mg caffeic acid/100 g), gooseberries (35.5 mg caffeic acid/100 g), and coffee (84 mg caffeic acid/100 g) are excellent sources of caffeic acid [7].

The consumption of caffeic-acid-containing foodstuff, including potatoes [8] and coffee [9], has been showed to be a protective factor against IBD. Caffeic acid can reach concentrations above 50 µmol/L (up to 126 µM) in the colon [10,11] where it can exert its biological effects. The number of studies regarding the beneficial effects of caffeic acid at the intestinal level is scant. Dextran sulfate sodium (DSS)-induced colitis mice models showed attenuation of intestinal inflammation in animals fed caffeic-acid-enriched diets [12,13]. In vitro studies investigating the cellular and molecular mechanism by which caffeic acid exerts intestinal beneficial effects are limited and mainly focused on colon cancer cells [14,15,16,17]. Nevertheless, studies on the biological activity of caffeic acid using in vitro noncancerous cell models that could be more representative in a context of gut inflammation context are absent.

The fibroblasts of the colon are subepithelial cells involved in the modulation of the inflammatory response. The destruction of the epithelial barrier architecture during chronic inflammation allows direct contact between subepithelial cells and lumen content, including phenolic compounds [18]. Based on this, noncancerous cell models of human intestinal myofibroblasts offer an interesting tool to investigate the anti-inflammatory effect of phenolic compounds, including caffeic acid.

In this study, we hypothesized that caffeic acid might be (one of) the molecule(s) responsible of the anti-inflammatory effects at the intestinal level associated with the consumption of caffeic-acid-rich foods. To test our hypothesis, we have investigated the effects of this phenolic acid, using concentrations reported in vivo (50–10 µM), on IL-1β-stimulated myofibroblasts of the colon as a relevant human model of intestinal inflammation. Specifically, we tested the effects on cyclooxygenase (COX)-2 expression, PGE_2_ biosynthesis, and cytokines and chemokines formation (IL-8, MCP-1, and IL-6). We also studied the capacity of caffeic acid, to hinder the formation of the pro-inflammatory AGEs, to inhibit the ACE activity, and its antioxidant, reducing, and chelating activity (using traditional and more recent techniques).

## 2. Materials and Methods

### 2.1. Materials

Caffeic acid was supplied by Extrasynthese (Genay, France). The reagents purchased from SIGMA (Sigma Chemical Co., St. Louis, MO, USA) used in the different experiments were as follows: (i) assays with cells: interleukin (IL)-1β, sodium pyruvate, Trypsin-EDTA (0.25%–0.03%), sodium bicarbonate, L-glutamine, Eagle’s minimum essential medium (EMEM), 3-(4,5-dimethylthiazol-2-yl)-2,5-diphenyltetrazolium (MTT), nonessential amino acids, fetal bovine serum (FBS), and antibiotics (i.e., penicillin/streptomycin); the antioxidant assays: 3-(2-pyridyl)-5,6-diphenyl-1,2,4-triazine-*p,p*′-disulfonic acid monosodium salt hydrate (ferrozine), 6-hydroxy-2,5,7,8-tetramethylchroman-2-carboxylic acid (Trolox), 2,2-diphenyl-1-picrylhydrazyl (DPPH); (ii) the antiglycative assays: methylglyoxal (MGO), bovine serum albumin (BSA), D-glucose; (iii) the angiotensin-converting-enzyme (ACE) assay: captopril, ACE isolated from porcine kidneys (EC 3.4.15.1). The reagent obtained from BACHEM (Bubendorf, Switzerland) was o-aminobenzoylglycyl-p-nitorphenylalanylproline (Abz-Gly-Phe(NO_2_)-Pro). MERK Millipore (Darmstadt, Germany) provided DMSO. A Millipore Direct-Q UV 3 System (Bedford, MA, USA) was used to obtain the ultrapure water used. Analytical-grade chemicals were used in all the experiments of this study.

### 2.2. Cell Culture Conditions

The myofibroblasts-like cell line CCD-18Co (ATCC^®^ CRL-1459) was purchased from the ATCC (American Type Culture Collection; Rockville, MD, USA). This cell line was grown and maintained following the instructions given by the manufacturer. Unless otherwise stated, EMEM (pH 7.2–7.4) was supplemented with 10% FBS (*v*/*v*), antibiotics (100 U mL^−1^ penicillin and 100 mg mL^−1^ streptomycin), 1.5 g/L sodium bicarbonate, 1 mM sodium pyruvate, 0.1 mM nonessential amino acids, and 2 mM L-glutamine. The cells were seeded at 6000 cells/cm^2^ in T75-flasks and incubated under optimal grown conditions (i.e., 37 °C, 5% CO_2_/95% air atmosphere, and constant humidity) for several days. Once ≥80% confluence was reached, the cells were subcultured. The passages and population doubling levels used for the experiments were 15–16 and 33–36, respectively.

### 2.3. Cell Viability Assay

We initially determined whether caffeic acid, under the conditions of our study, showed cytotoxic effects on the myofibroblasts. A 10 mM stock solution of caffeic acid was prepared in DMSO. The cells were treated with 1 ng/mL IL-1β together with 50 μM caffeic acid (0.5% DMSO, *v*/*v* final concentration) or in absence of this phenolic acid for 24 h. DMSO (0.5%, *v*/*v*)-treated cells were used as controls. Next, the cells were washed with PBS and incubated with 1 mg/mL MTT (prepared in FBS-deprived EMEM) for 4 h to allow its conversion to formazan. Following the 4 h incubation, the MTT solution was removed, the formazan diluted in DMSO and the absorbance (570 and 690 nm as test and reference wavelengths, respectively) determined using a microplate reader (SynergyH1, BioTek, Winooski, VT, USA). The effect of caffeic acid on cell viability was calculated from three different replicates (*n* = 3). Each replicate was analyzed in triplicates.

### 2.4. Measurement of PGE_2_ Biosynthesis in IL-1β-Stimulated Myofibroblasts by ELISA

Next, we tested the capacity of caffeic acid to modulate the formation of PGE_2_ in IL-1β-stimulated myofibroblasts of the colon. In brief, confluent cells grown in 96-well plates were incubated in FBS-deprived medium for 12 h, and then co-treated with 1 ng/mL IL-1β and caffeic acid (0.5% DMSO, *v*/*v*) at 50, 10, and 1 µM for 24 h. In absence of caffeic acid, the myofibroblasts were treated with DMSO (0.5% *v*/*v* final concentration) in the presence or absence of 1 ng/mL IL-1β. The culture medium was frozen (−80 °C) and later analyzed. The concentration of PGE_2_ in culture medium was measured at 405 nm using an absorbance-detecting microplate reader (Infinite M1000 Pro, Tecan, Grodig, Austria). The ELISA kit was provided by Cayman (San Diego, CA, USA). The effect of caffeic acid on PGE_2_ was calculated from three different replicates (*n* = 3). Each replicate was analyzed in triplicate.

### 2.5. Measurement of IL-1β-Induced COX-2 Expression by Western Blot

After the collection of the culture medium for the analysis of PGE_2_ (see above), we added ice-cold Radioimmunoprecipitation assay (RIPA) buffer with protease inhibitors to the cells and extracted the cellular protein as described elsewhere [19]. The protein analyzed by Western blot was quantified (DC colorimetric assay, Bio-Rad, Warsaw, Poland), loaded in SDS-PAGE at 10% (20 µg of protein in each lane), and separated. Next, it was transferred to PVDF membranes, which were blocked (5% albumin, w/v), and then incubated with primary (COX-2, 1:1000 and Glyceraldehyde 3-phosphate dehydrogenase -GAPDH-, 1:2500; Cayman, San Diego, CA, USA) and secondary (goat anti-mouse and anti-rabbit, 1:10,000; LI-COR antibody, Lincoln, NE, USA) antibodies using the same conditions previously described elsewhere [19]. Odyssey Infrared Image System v. 1.2 (Li-COR Bioscience, Lincoln, NE, USA) allowed the detection and quantification of the proteins. The effect of caffeic acid on COX-2 expression was determined from three different replicates (*n* = 3).

### 2.6. Measurement of Cytokine Production in IL-1β-Stimulated Myofibroblasts by ELISA

Based on previous studies [20,21], the culture medium was also used for the analysis of further molecules involved in the inflammatory response, including chemokines (i.e., IL-8 and MCP-1) and cytokines (such as IL-6). PeproTech (Rocky Hill, NJ, USA) provided the ELISA kits. The concentration of these cytokines was measured at 405 and 650 nm (test and reference wavelengths, respectively) using an absorbance-detecting microplate reader (Infinite M1000 Pro, Tecan, Grodig, Austria). The effect of caffeic acid on these molecules was calculated from 3 different replicates (*n* = 3). Each replicate was analyzed in triplicates.

### 2.7. Inhibition of the Formation of the Advanced Glycation End Products (AGEs)

Stock solutions at 1 mM of caffeic acid were prepared (DMSO/0.1 M phosphate buffer at pH 7.4; 1:99; *v*/*v*) to study its anti-AGE activity. For comparative purposes, L-ascorbic and uric acid as well as the positive control aminoguanidine (AG) [22] were prepared at the same concentration using the same buffer. The inhibitory activity was tested using the BSA/glucose and BSA/MGO systems as reported elsewhere [19]. The fluorescent intensities for each system were 330 nm excitation and 410 nm emission for the BSA–glucose system and 340 nm excitation and 420 nm emission for the BSA/MGO system. The percentage (%) of inhibition was obtained from nine repetitions (*n* = 9).

### 2.8. Angiotensin-I-Converting Enzyme Inhibitory Assay

The inhibition of ACE activity by caffeic acid (compared to GSH and L-ascorbic and uric acid) was determined following the procedure reported by Sentandreu and Toldrá [23]. A wide range of concentrations of each compound (dissolved in 70% methanol) was prepared by serial dilutions in deionized water to determine the half-maximal inhibitory concentration (IC_50_) value (calculated by linear regression analysis of logarithmic plots). The emitted fluorescence by the hydrolysis reaction of the substrate Abz-Gly-Phe(NO_2_)-Pro by the enzyme was measured using a multiskan microplate fluorometer at an excitation of 365 nm and an emission of 405 nm. The percentage of inhibition was calculated as follows:
Relative ACE activity % = 100 − (ΔRFU_sample_ × 100/ΔRFU_negative control_)(1)
where ΔRFU = RFU_at time 30_ − RFU_at time 0_.

The IC_50_ values were determined from three different experiments.

### 2.9. Antioxidant Activity

Stock solutions at 1 mM caffeic acid were prepared in 50% methanol (*v*/*v*), and the concentration was verified according to previous studies [24]. L-ascorbic and uric acid stock solutions were prepared following the same procedure. The DPPH-radical-scavenging activity (DPPH RSA) [25] of these compounds was tested under Tª-controlled conditions using a spectrophotometer UV-1601PC with CPS-Controller (Shimadzu Corp., Kyoto, Japan). The reaction mixture was incubated in the dark at room temperature for 20 min, after which the absorbance at 517 nm was recorded. The results of this assay were calculated considering the data obtained from nine independent assays (*n* = 9) and showed as Trolox equivalents (mM).

### 2.10. Reducing Activity

To investigate its reducing activity, 500 µM caffeic acid prepared in 50% (*v*/*v*) methanol was mixed with and equal volume of 0.1 M Britton–Robinson buffer (pH 6.0). L-ascorbic and uric acid were prepared under the same conditions. As described by Zielińska et al. [26], cyclic voltammetry (CV) was determined using a G750 Gamry potentiostat (Warminster, PA, USA). The potential ranges for Trolox and the investigated compounds were –0.1 to 1.3 V and –0.1 to 1.2 V, respectively at a scan rate of 0.1 V s^−1^. The results of this assay were calculated considering the data obtained from nine independent assays (*n* = 9) and showed as Trolox equivalents (mM).

### 2.11. Ferric-Reducing/Antioxidant Power (FRAP) Activity

The FRAP assay was carried out according to previous investigations [27]. Caffeic acid, L-ascorbic, and uric acid were prepared as described above. A standard curve of Trolox (0.034–0.612 mM) was used to quantify FRAP activity. The absorbance of the reagent mixture was measured at 593 nm after 30 min incubation at 25 °C. The results of this assay were calculated considering the data obtained from nine independent assays (*n* = 9) and showed as Trolox equivalents (mM).

### 2.12. Chelating Activity

To determine the chelating activity, a previously described procedure [28] was followed. The Fe^2+^ ions’ standard curve (5 to 60 μM) was used for quantification. The absorbance of the ferrous ion–ferrozine complex was measured at 562 nm. The chelating activity showed as percentage (%) was calculated based on the results obtained from nine repetitions (*n* = 9).

### 2.13. Statistical Analysis

The results of the chemical and cellular/molecular assays were obtained from nine (*n* = 9) and three (*n* = 3) independent assays, respectively. The statistical analysis used with the normal distributed data of the different assays was one-way analysis of variance (ANOVA) followed by Dunnet’s post hoc test. Significant differences are indicated in the figures as follows: * *p* < 0.05, ** *p* < 0.01, and *** *p* < 0.001. Furthermore, the correlation between the parameters investigated was tested using the Pearson test. Significant differences were analyzed using Prism 5 (GraphPad, La Jolla, CA, USA). Graphics and figures were prepared using Sigma Plot 13.0 (Systat Software, San Jose, CA, USA) and ChemDraw Professional v. 16.0.1.4 (Perkin Elmer Informatics Inc., Cambrige, MA, USA).

## 3. Results

### 3.1. Caffeic Acid Exerted No Cytotoxic Effects

Under the conditions of the assays, the cell viability of the myofibroblasts cotreated with IL-1β and caffeic acid was similar to that observed in control cells (above 90%), indicating a lack of cytotoxic effects (Appendix A).

### 3.2. Caffeic Acid Ameliorates the Effect of IL-1β on the Biosynthesis of PGE in Myofibroblasts of the Colon, CCD-18Co

We first tested whether caffeic acid, at nontoxic concentrations, reduced the effect of IL-1β (1 ng/mL) on the formation of PGE_2_ in myofibroblasts of the colon. As showed in Figure 2, caffeic acid reduced the formation of PGE_2_ in the presence of IL-1β in a dose-dependent manner. Thus, the increase of PGE_2_ observed in IL-1β-treated myofibroblasts (*p* < 0.001) was ameliorated in the presence of caffeic acid at 50 and 10 µM (*p* < 0.05), whereas at 1 µM the reduction (~26%) observed was not significant.

### 3.3. Caffeic Acid Downregulates the Expression of COX-2 in IL-1β-Stimulated Myofibroblasts of the Colon, CCD-18Co

We next determined whether the effect of caffeic acid on PGE_2_ was associated with a downregulation of COX-2. In this assay, we only tested those concentrations of caffeic acid that showed significant effects (*p* < 0.05) on PGE_2_. As showed in Figure 3, 1 ng/mL IL-1β upregulated (*p* < 0.001) the expression of COX-2 in the myofibroblasts compared with the untreated cells. The cells co-treated with IL-1β and caffeic acid showed a significant downregulation of COX-2 compared with the IL-1b-treated cells at 50 and 10 µM (*p* < 0.05).

### 3.4. Caffeic Acid Lacked the Capacity to Modulate the Formation of Chemokines (IL-8 and MCP-1) and Cytokines (IL-6) in IL-1β-Stimulated Myofibroblasts of the Colon, CCD-18Co

We next tested whether caffeic acid, at 50 and 10 µM, was able to modulate the biosynthesis of other molecules involved in the inflammatory response, including IL-8, MCP-1, and IL-6. Our results showed increased level of the cytokines investigated in the presence of 1 ng/mL IL-1β (Figure 4). Only a small reduction (~27%) of the concentration of IL-8 was observed in the presence of caffeic acid at 50 and 10 µM, although this effect was only significant at 50 µM. The concentration of MCP-1 and IL-6 remained unaffected in the cells co-treated with IL-1β and caffeic acid compared with the stimulated cells (Figure 4B,C).

### 3.5. Inhibition of the Formation of the Advanced Glycation End Products (AGEs)

The inhibitory effect caffeic acid against AGE formation is shown in Figure 5. As expected, a considerable inhibition (higher than 70% in both models) of AGEs formation was observed in the presence of the positive control, AG. In the presence of caffeic acid, our results showed a dose-dependent inhibition (from 0.01 to 1.0 mM) on the formation of the AGEs in the BSA/glucose system (y = 12.50x + 53.687; R^2^ = 0.81) and in the BSA/MGO system (y = 15.44x + 43.75; R^2^ = 0.99), reaching higher inhibitory values than those of the positive control. The anti-AGE activity of ascorbic acid and uric acid were lower than those for caffeic acid. The ranking of anti-AGE activity was caffeic acid (CFA) > ascorbic acid (AA) > uric acid (UA) > aminoguanidine (AG). These results clearly indicate that caffeic acid displays a strong inhibitory capacity on the AGE formation.

### 3.6. ACE Inhibitory Assay

The results obtained for the ACE inhibitory activity of caffeic acid in comparison to that of captopril, glutathione (GSH), and L-ascorbic acid are shown in Table 1. Caffeic acid showed a very low ACE inhibitory activity. In contrast, captopril, which is a synthetic drug, showed extremely high activity. The ACE inhibitory activity of GSH, a small tripeptide, was about 126-fold higher than that determined for caffeic acid. The lowest ACE inhibitory activity was noted for L-ascorbic acid. The ranking of ACE inhibitory activity was as follows: captopril > GSH > caffeic acid > ascorbic acid. These results clearly indicate that caffeic acid displays a very low ACE inhibitory activity especially when compared to that provided by GSH.

### 3.7. The Antioxidant, Reducing, and Chelating Activity of Caffeic Acid

Caffeic acid exhibited antioxidant activity on the comparable level to that of L-ascorbic acid. The DPPH RSA values for both compounds were only lower by 25% than the radical scavenging activity of uric acid. The ranking of the FRAP values was as follows: uric acid > L-ascorbic acid > caffeic acid. In contrast, caffeic acid showed a very high chelating activity, which was 42-fold higher than that of ascorbic acid and 76-fold higher than that of uric acid.

### 3.8. Reducing Power of Caffeic Acid by CV

The CV method provided information about the reducing activity of caffeic acid in comparison to L-ascorbic acid and uric acid (Figure 6).

The total reducing activity results from the combination of the oxidation potential (*E_pa_*) and the intensity of the anodic current (*I*a). The area under the anodic wave (associated with the total charge) is the determined to show the reducing activity of the sample. The area under the anodic current waveform obtained for CFA, UA, AA, and Trolox was calculated, and the reducing activity was expressed as Trolox equivalents in mM (Table 2).

The reducing power of caffeic acid was lower by 20% as compared to that of ascorbic and uric acids (Table 2). To characterize the reducing activity of caffeic acid compared to AA and UA, E_pa_ values were used (low values indicate strong reducing power). As shown in Figure 6, the oxidation peak potentials were as follows: uric acid (0.621 V) > Trolox (0.411 V) > L-ascorbic acid (0.343 V) > caffeic acid (0.334 V). These results follow a similar pattern to that observed in the FRAP assay, what is reflected in the positive correlation (*r* = 0.99) found between both assays. Conversely, regarding the chelating activity, the correlation observed was negative (*r* = −0.99).

## 4. Discussion

In 2009, studies by Ye et al. [12] described the amelioration of DSS-induced intestinal inflammation in mice fed a caffeic-acid-enriched diet. These beneficial effects included protection of the intestinal architecture, reduction of myeloperoxidase activity, and the modulation of inflammatory markers [12]. In 2016, the study by Zhang et al. [13] described further mechanisms of attenuation of colitis by caffeic acid, including modulation of gut microbiota and inhibition of the nuclear factor kappa-light-chain-enhancer of activated B cells (NF-kB) pathway together with the reduction of cytokine formation and immune cell infiltration [13]. Even though these studies provide evidence of the anti-ulcerative effects of caffeic acid, our knowledge of its underlying molecular mechanisms is limited. Here, we show that caffeic acid, at concentrations detected in the colon [10,11], could exert anti-inflammatory effects through the inhibition of the COX-2 and PGE_2_ as well as the biosynthesis of chemokines (such as IL-8) in a human intestinal cell model of inflammation. Furthermore, we also describe the capacity of caffeic acid to inhibit the formation of AGEs (related to its chelating activity), which can act as pro-inflammatory compounds at the intestinal level.

Both COX-2 and its product PGE_2_ are targets of considerable importance in the treatment of intestinal inflammation [29]. PGE_2_ is a critical player in the maintenance of the intestinal barrier function, normal epithelial growth, and in the regulation of the immune system under physiological conditions [30]. However, it has been reported that an abnormal, high level of PGE_2_ promotes intestinal damage and colorectal cancer [31]. Thus, the modulation of the PGE_2_ biosynthesis is an essential strategy to avoid its deleterious effects at the intestinal level. The capacity of caffeic acid to reduce PGE_2_ levels has been tested (at concentrations from 25 to 400 µM) in immune cells such as LPS-stimulated macrophages [32,33]. Here, we describe, for the first time, that caffeic acid at relevant concentrations shows capacity to counteract the formation PGE_2_ in a model of human intestinal cells treated with the pro-inflammatory cytokine IL-1β. This effect was related to the lower COX-2 expression detected in the myofibroblasts co-treated with caffeic acid and IL-1β compared with that in the inflamed cells. Based on previous studies, the effect of caffeic acid on COX-2 and PGE_2_ most likely occurred via inhibition of the transcription factor NF-kB [13,34], although its interaction with receptors (i.e., toll-like receptors) involved in the activation of this route has also to be considered [35,36].

Inflammation is a complex response modulated by a plethora of molecules. In addition to PGE_2_, IL-8, IL-6, and MCP-1 also play a critical role in the development of intestinal inflammation [4]. IL-8 is a chemokine biosynthesized by a variety of cells of the gastrointestinal tract such as myofibroblasts in response to pro-inflammatory cytokines [21] promoting the recruitment of immune cells to the inflammation site. Sustained expression as well as high levels of IL-8 have been documented in patients suffering gastrointestinal inflammation and cancer [37], which justifies the study of compounds targeting the biosynthesis of this molecule. Information on the capacity of caffeic acid to attenuate the formation of IL-8 in human intestinal cells is scant. In vitro studies have described a reduction of IL-8 in H_2_O_2_- and TNF-α-treated Caco-2 cells at concentrations from 0.25 to 2 mM [16,17]. Similarly, our results showed that lower concentrations (10 and 50 µM) of caffeic acid exerted a reduction of IL-8 in the IL-1β-stimulated myofibroblasts of the colon. These results suggest that caffeic acid targets IL-8 biosynthesis in epithelial cells (i.e., Caco-2) and myofibroblasts of the colon (i.e., CCD-18Co) during intestinal inflammation. Considering the stimulating effect of PGE_2_ on IL-8 formation in fibroblasts [38,39], a possible explanation for this effect could be found in the inhibitory effect of caffeic acid on COX-2 and PGE_2_ described above. Unlike IL-8, caffeic acid lacked the capacity to reduce the IL-1β-induced biosynthesis of IL-6 and MCP-1. These results are in contrast with those reported in other cells lines such as THP-1-derived macrophages [40]. This suggests that caffeic acid can modulate the inflammatory response in a range of different cell types at the intestinal level, highlighting the importance of the study of different cell lines.

An additional mechanism by which caffeic acid might exert benefits against IBD is the inhibition of ACE activity. New insights have found a relation between the inhibition of ACE and alleviation of IBD severity in humans and animal models [41,42]. Synthetic compounds such as captopril as well as different phenolic compounds (including caffeic acid derivatives) have been shown to be efficient ACE inhibitors [43,44]. Our analyses indicate that caffeic acid exerts a weak ACE inhibition (1 × 10^6^-fold lower than captopril), suggesting that an improvement against intestinal inflammation exerted by caffeic acid via this mechanism is less likely.

The consumption of diets rich in processed food, which contain a high level of dietary AGEs, is related to increased colonic oxidative stress and inflammation [45]. The reduction of dietary AGEs is an interesting therapeutic target for the prevention/treatment of IBD. In this study, we used two different models to test the inhibitory capacity of caffeic acid (compared with AA, UA, and the positive control, AG) against AGE formation. Caffeic acid was the most effective molecule exerting a dose-dependent antiglycative effect in both models. An explanation for this effect is related to the MGO-trapping capacity of caffeic acid [46], which might be exerted via the protection of lysine, arginine, and/or histidine groups [47]. This mechanism of protection is rather common to ortho-diphenolic compounds with a similar structure to that of caffeic acid and is linked to the antioxidant and chelating activity [19,47].

Despite the abundant investigations regarding the antioxidant properties of caffeic acid, there is a lack of consensus about its antioxidant mechanism [48]. In agreement with previous studies [49], caffeic acid exhibited the lowest antioxidant activity of the molecules tested, which could be related to the low number of -OH groups [50]. Despite the low number of -OH moieties, it was the most effective molecule regarding the chelation of Fe (II) ions. A likely mechanism could be the binding of the iron ion to the catechol moiety [51], which is absent in the uric and ascorbic acid. These results suggest that caffeic acid through a chelating–antioxidant mechanism might prevent the formation of AGEs, inhibiting the reaction between hydroxyl radical and target molecules (i.e., MGO). Additionally, caffeic acid might prevent oxidative damage related to iron (and other transition metals) in the gastrointestinal tract, where it is highly absorbed, reducing the formation of Fenton-generated hydroxyl radicals and their harmful effects regarding intestinal inflammation.

## 5. Conclusions

Overall, this study shed some light about the chemistry of caffeic acid and the molecular mechanisms associated with its anti-inflammatory effects at the intestinal level. So far, in vivo studies have described beneficial effects associated with the consumption of caffeic-acid-enriched diets against induced colitis [12,13]. The results described in the in vitro models provide additional information about the mechanisms by which this phenolic compound may exert its beneficial effects against colitis and its undesirable effects (i.e., ulcerative complications). However, more investigation (preclinical and clinical studies) is necessary to unequivocally prove that caffeic acid is (one of) the molecule(s) responsible of the effects observed in vivo after the consumption of hydroxycinnamic-acid-containing food. Thus, some aspects that deserve attention in future studies are as follows:

New in vitro studies using a range of relevant cell lines (with different features) and correctly designed [52] will be essential to enlarge our knowledge about further mechanisms of action. Based on the reported effects of caffeic acid on the well-defined 5-lypoxygenase (5-LOX) [53] and COX-2 (see above) pathways, an interesting approach could be the study of its effects on the formation of the more recently described 5-LOX/COX-2 hemiketal (HK) eicosanoids, HKE_2_ and HKD_2_ [54,55,56,57,58].

Considering the limited number of in vivo studies, more investigation using relevant animal models is required.

There is limited evidence from human studies regarding the beneficial effects of caffeic acid and(or) caffeic-acid-rich foodstuff. A higher number of correctly designed human studies is necessary.

Based on the role of gut microbiota as a key player of the biological activity of phenolic compounds [59,60], the study of the “caffeic acid–gut microbiota” interaction is essential.

## Figures and Tables

**Figure 1 nutrients-13-00554-f001:**
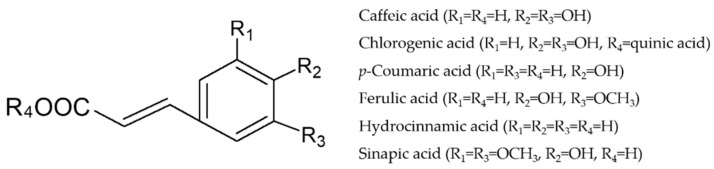
Chemical structure of caffeic acid and related hydroxycinnamic acids.

**Figure 2 nutrients-13-00554-f002:**
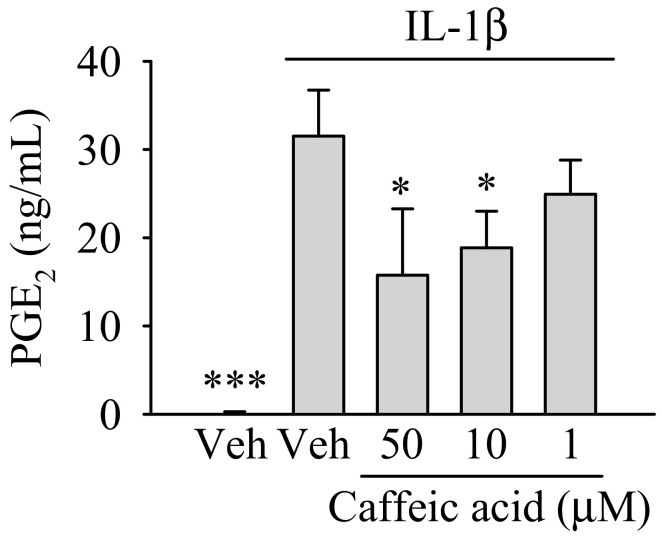
Dose-dependent effect of caffeic acid on the biosynthesis of prostaglandin E_2_ (PGE_2_) in IL-1β-treated myofibroblasts of the colon, CCD-18Co. In the absence of caffeic acid, the cells were treated with 0.5% (*v*/*v*) DMSO in the presence or absence of IL-1β (inflamed or control cells, respectively). The bar graph shows the data as the average ± standard deviation (SD) of three independent experiments (*n* = 3). Each treatment was performed in triplicate. Significant differences are indicated in the figures as follows: * *p* < 0.05 and *** *p* < 0.001.

**Figure 3 nutrients-13-00554-f003:**
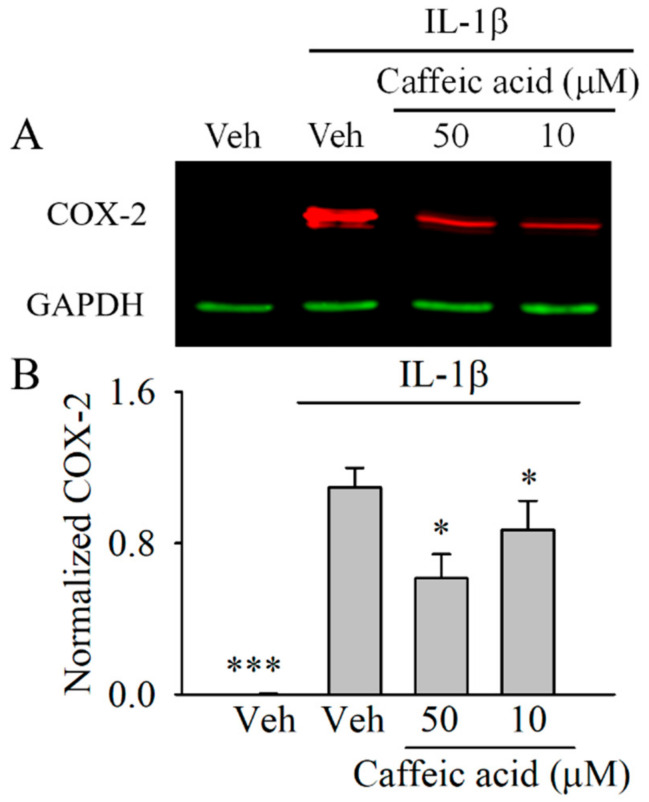
Dose-dependent effect of caffeic acid on cyclooxygenase 2 (COX-2) expression in interleukin 1 beta (IL-1β)-treated myofibroblasts of the colon, CCD-18Co. Cells treated with DMSO (0.5% *v*/*v*) were used as control. In the absence of caffeic acid, the cells were treated with 0.5% (*v*/*v*) DMSO in the presence or absence of IL-1β (inflamed or control cells, respectively). The expression of COX-2 was normalized to Glyceraldehyde 3-phosphate dehydrogenase –GAPDH- (ratio COX-2/GAPDH) and shown as normalized COX-2. (**A**) Illustration of the effect of 50 and 10 µM caffeic acid on the IL-1β-induced COX-2 expression in the myofibroblasts of the colon, CCD-18Co. (**B**) The bar graph shows the data as the average ± standard deviation (SD) of three independent experiments (*n* = 3). Significant differences are indicated in the figures as follows: * *p* < 0.05 and *** *p* < 0.001.

**Figure 4 nutrients-13-00554-f004:**
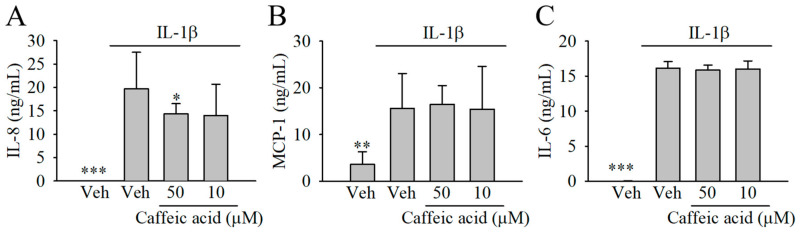
Effect of caffeic acid on the biosynthesis of (**A**) IL-8, (**B**) monocyte chemoattractant protein-1 (MCP-1), and (**C**) IL-6 in IL-1β-stimulated myofibroblasts of the colon, CCD-18Co. In the absence of caffeic acid, the cells were treated with 0.5% (*v*/*v*) DMSO in the presence or absence of IL-1β (inflamed or control cells, respectively). The bar graph shows the data as the average ± standard deviation (SD) of three independent experiments (*n* = 3). Each treatment was performed in triplicate. Significant differences are indicated in the figures as follows: * *p* < 0.05, ** *p* < 0.01, and *** *p* < 0.001.

**Figure 5 nutrients-13-00554-f005:**
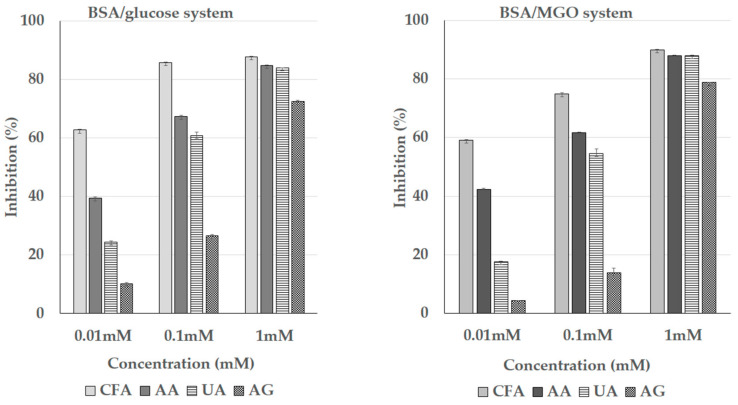
The concentration-dependent inhibitory effects of caffeic acid (CFA), ascorbic acid (AA), uric acid (UA), and aminoguanidine (AG) against advanced glycation end product (AGE) formation as measured in bovine serum albumin (BSA)/glucose and BSA/methylglyoxal (MGO) assays.

**Figure 6 nutrients-13-00554-f006:**
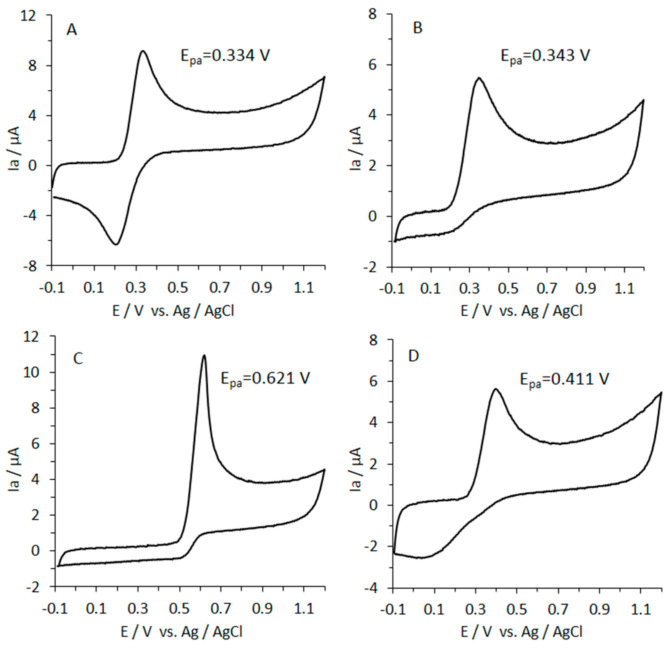
Cyclic voltammograms of 0.25 mM of standards solution (final concentration) of caffeic acid (**A**), L-ascorbic acid (**B**), uric acid (**C**), and Trolox (**D**) in 0.1 Britton–Robinson (B–R) buffer (pH 6.0) recorded from −100 to +1200 mV; scan rate 100 mV s^−1^.

**Table 1 nutrients-13-00554-t001:** The half-maximal inhibitory concentration (IC_50_) of caffeic acid in comparison with captopril and blood plasma antioxidants for angiotensin-converting enzyme (ACE) inhibition.

Standard	IC_50_ (µg/mL)	Linear Regression
Caffeic acid	1616.78 ± 1.45 ^c^	y = 0.021x + 15.401
Captopril	0.0013 ± 0.001 ^a^	y = 23,075x + 20.606
GSH	12.79 ± 1.70 ^b^	y = 0.435x + 44.431
L-ascorbic acid	14,320.09 ± 5.10 ^d^	y = 0.003x + 4.176

Data are expressed as means ± S.D. of nine independent assays (*n* = 9). Different letters in a column means significant differences (*p* < 0.05). GSH, glutathione.

**Table 2 nutrients-13-00554-t002:** The antioxidant, reducing, and chelating activity of caffeic acid vs. L-ascorbic and uric acids.

Compound/Assay	Antioxidant Activity (mM Trolox)	Reducing Activity (mM Trolox)		Chelating Activity (%)
	DPPH RSA	FRAP	CV	FZ
Caffeic acid	0.90 ± 0.01 ^a^	0.89 ± 0.01 ^a^	0.96 ± 0.01 ^a^	75.80 ± 1.07 ^c^
L-ascorbic acid	0.94 ± 0.01 ^a^	1.35 ± 0.02 ^b^	1.18 ± 0.02 ^b^	1.79 ± 0.36 ^b^
Uric acid	1.18 ± 0.01 ^b^	1.42 ± 0.03 ^c^	1.21 ± 0.02 ^b^	0.99 ± 0.11 ^a^

Results were provided by DPPH-radical-scavenging activity (DPPH RSA), ferric-reducing/antioxidant power (FRAP), cyclic voltammetry (CV), and ferrozine (FZ). Data are expressed as means ± S.D. obtained from nine different/independent assays (*n* = 9). Different letters in the same column mean significant differences regarding a respective experiment (*p* < 0.05).

## Data Availability

Data is contained within the article or Appendix A.

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
