# Peer review of "Caffeic Acid Modulates Processes Associated with Intestinal Inflammation"

_nutrients, 2021, doi:10.3390/nu13020554_

Round 1
Reviewer 1 Report
Summary
This study observed the effect of caffeic acid on COX2/PGE2 pathway, chemokines and cytokine production, the formation of AGEs, and antioxidant ability using IL-1b-treated human myofibroblast of colon cell line. The author revealed that caffeic acid targets COX2/PGE2 pathway, reduces AGEs formation via chelating activity, and down-regulates IL-8 levels. However, the explanations about the association between parameters observed in this study is needed.
Comments
- Abstract section:
There is a wrong spelling (myofibrobast). Please fix it.
- Introduction section:
The author mentioned that the biological effects of caffeic acid were mainly focused on colon cancer cells, therefore, study using in vitro non-cancerous cell model is needed. However, myofibroblast of colon was used, not colon epithelial cells. What is the rational to choose this cell as a model for the study. Is there any evidence showing that myofibroblasts associated with intestinal inflammation or with colon epithelial cells in intestinal inflammation?
This point might be added in an introduction part for more understanding.
- Material and methods section:
3.1 In material section: the author use (i) to identify the chemical used in each assay, however, it is followed in (ii), (iii), etc. This part should be unified.
3.2 As 1 and 10 µM caffeic acid are also used for this study, why the author chose only 50 µM for viability test?
3.3 Why the author chose only 1 µM for Inhibition of AGEs and antioxidant activity tests?
3.4 The author mentioned that the concentration of caffeic acid can reach maximum level at 126 µM in colon. Why excessive concentration of caffeic acid (500 µM) was used for reducing activity test?
- Results section:
4.1 In the result section, the author mentioned that cell viability was above 90% but there is no support result. It might be better to also put the viability result in supplementary data.
4.2 Are IL-8, IL-6, and MCP-1 major or specific cytokines and chemokines produced by myofibroblasts in response to IL-1b activation, why the author chose to determine the change of these cytokines upon caffeic acid treatment? Is there any correlation between COX2-PGE2 pathway with these cytokine levels?
4.3 The result in section 3.4 showed the reduction of IL-8 levels in caffeic acid treatment at 50 and 100 µM (x and x, respectively). What are the x and x?
4.4 The result in section 3.5 refers to wrong figure (it should be figure 5 not figure 4) and in section 3.8 refers to wrong figure (it should be figure 5 not figure 6)
4.5 The author has determined each effect of caffeic acid, however, there is no explanation why each parameter was observed. Are these effects associated each other, for examples, the increase the inhibition of AGE formation might lead to the reduction of cytokine production?
4.6 What are the interpretations that caffeic acid has highest chelating activity but low antioxidant activity as well as reducing activity?
4.7 This study showed that caffeic acid can reduce the levels of PGE2 and IL-8 upon IL-1b activation which might be related with an alteration of NF-kb pathways. Is there any evidence supports that NF-kb pathway is also inhibited by caffeic acid?
Reviewer 2 Report
This manuscript is an informative report to show “Caffeic acid modulates processes associated with intestinal inflammation”, however, more informative discussion for the following doubtful points are recommended.
- You mentioned the aim was to study the capacity of caffeic acid, at plausible concentrations from an in vivo point of view, to modulate mechanisms related to intestinal inflammation.
A) The study was designed with the myofibroblasts-like cell line CCD-18Co, not the primary cell line, what does “an in vivo point of view” means?
B) Which result could explain the plausible concentrations from an in vivo point of view, to modulate mechanisms related to intestinal inflammation? If youfind it, you have to mention it in the conclusion part.
C) In the discussion, you want to find the molecular mechanism of the anti-ulcerative effects of caffeic acids. I’d like to talk you focus on the anti-inflammatory effects of caffeic acids as one of the mechanisms using your well-designed in vitro.
- For the hypothesis-methodology process,
A) In the last paragraph of the introduction part, you have to mention the purposes. The readers wonder; what mechanisms could be focused on the hypothesis, or how the intestinal inflammation connects with anti-oxidation, AGE formation, or ACE inhibition. First of all, what mechanisms are suggested for your target, anti-ulcerative effects of caffeic acids, and what was your hypothesis in “Introduction”, and which methodology will be the best for them in ‘Methods”, eventually, which metabolites were related to the mechanism in “Discussion”. The connecting system of “hypothesis to prove the mechanisms-methodology-results” was not be cleared.
B) Since I wonder the ACE could be significantly modulated in the Intestine Cells, despite the benefits of caffeic acids in IBD, you need to discuss with “Cell Reference”.
C) To prove the antioxidant effect of polyphenolic acids, various methods, such as DPPH radical scavenging, cyclic voltammetry (CV), FRAP activity, and chelating activity, were tested. Since their antioxidant effects were not significant compared to the control in most methods except chelating activity, you should have discussed why they were inconsistent. Without the discussion, you decide whether the antioxidant effect of caffeic acids may be related, may be not related, to one of mechanisms of intestinal inflammation.
- Minor revisions
A) Fig 5 and Fig 6 were wrongly entered in the text.
B) You would better explain the results of cyclic voltammetry in detail.
Reviewer 3 Report
This article provides some insights into anti-inflammatory effects of caffeic acid (CA) in colon. Based on their findings, the Authors consider the potential use of CA to act against gut inflammation. However, the results presented by the Authors might be stronger with more cell lines included into study. The Materials and Methods section should be improved.
This manuscript presents interesting findings that may be attractive for the Readers. Yet, some revisions are needed
The introduction provides sufficient background and includes relevant references.
Following aspect of the paper need to be reconsidered:
Materials and Methods
It would be appropriate to describe the methods more extensively, since the existing information given in the section is scarce.
In particular:
The Authors should explain what is the rationale to use only one cell line in this study. This must be discussed.
Please specify the particular numbers of strains used according to ATCC designation
Please, give names of manufacturers of media/sera were used for cell line
The Authors must clarify what techniques were used in their experiments (e.g. colorimetric assay, western blott analysis;
more specific information is needed e.g “the concentration of … was measured spectrophotometrically using microplate reader (type, company, country) at wavelength … nm, with reference wavelength at… nm)”
Please specify the wavelength used for all spectrophotometric assays (e.g 2.9. Antioxidant Activity)
Please specify the source of antibody.
The amount of protein in each lane (western blot) should be listed
Th Authors may reconsider the captions of the sections 2.4-2.6 (the existing might be rather used in Results section)
Results
3.1. Caffeic acid exerted no cytotoxic effects
Why the Authors did not included a figure showing the cytotoxicity of CA (and possibly other compounds tested in their experiments) in the MS or in supplementary data?
Figures and tables
Please, elucidate briefly why the Authors used linear regression to assess IC 50? (Table 1)
Discussion
The Authors presented interesting data how CA modulate some important proteins associated with intestinal inflammation. However, I am not sure if, based on their findings, the Authors may discuss the effect of CA on the whole COX-2/PGE2 pathway – then the effect on some up- and downstream proteins and molecular targets triggered upon exposition of the cells to CA should be also presented.
“Increase the number of accurately designed clinical trials.” ? What do you mean?
“Based on the role of gut microbiota as a key player of the biological activity of phenolic compounds [49], the study of the interaction “caffeic acid-gut microbiota” is essential” – the Authors did not included study on gut microbiota interactions in their experiments – please, reconsider if this issue should be discussed here.
Minor remarks:
The English language style might be improved by a native speaker
The line numbering throughout the text is missing.
“In 2009, studies by Ye et al [please, add ref] described…”
The same: “In 2016, the study by Zhang et al. [please, add ref] described…”
In my opinion, minor revision is needed. After changes the manuscript is suitable for the publication.
Kind regards,
Round 2
Reviewer 1 Report
I have no more comments.
Reviewer 2 Report
Responses of authors are reasonable, although there were something to be misunderstood. However, I agree to accept this manuscript.